# Hemi- and Homozygous Loss-of-Function Mutations in DSG2 (Desmoglein-2) Cause Recessive Arrhythmogenic Cardiomyopathy with an Early Onset

**DOI:** 10.3390/ijms22073786

**Published:** 2021-04-06

**Authors:** Andreas Brodehl, Alexey Meshkov, Roman Myasnikov, Anna Kiseleva, Olga Kulikova, Bärbel Klauke, Evgeniia Sotnikova, Caroline Stanasiuk, Mikhail Divashuk, Greta Marie Pohl, Maria Kudryavtseva, Karin Klingel, Brenda Gerull, Anastasia Zharikova, Jan Gummert, Sergey Koretskiy, Stephan Schubert, Elena Mershina, Anna Gärtner, Polina Pilus, Kai Thorsten Laser, Valentin Sinitsyn, Sergey Boytsov, Oxana Drapkina, Hendrik Milting

**Affiliations:** 1Erich and Hanna Klessmann Institute, Heart and Diabetes Center NRW, University Hospital of the Ruhr–University Bochum, Georgstrasse 11, 32545 Bad Oeynhausen, Germany; bklauke@hdz-nrw.de (B.K.); cstanasiuk@hdz-nrw.de (C.S.); gpohl@hdz-nrw.de (G.M.P.); jgummert@hdz-nrw.de (J.G.); agaertner@hdz-nrw.de (A.G.); hmilting@hdz-nrw.de (H.M.); 2National Medical Research Center for Therapy and Preventive Medicine, Petroverigsky per., 10, bld. 3, 101000 Moscow, Russia; meshkov@lipidclinic.ru (A.M.); andorom@yandex.ru (R.M.); olgakulikova2014@mail.ru (O.K.); sotnikova.evgeniya@gmail.com (E.S.); divashuk@gmail.com (M.D.); kudryavtseva6041995@yandex.ru (M.K.); azharikova89@gmail.com (A.Z.); SKoretskiy@gnicpm.ru (S.K.); drapkina@bk.ru (O.D.); 3Cardiopathology, Institute for Pathology and Neuropathology, University Hospital Tuebingen, Liebemeister-strasse 8, 72076 Tuebingen, Germany; karin.klingel@med.uni-tuebingen.de; 4Comprehensive Heart Failure Center (CHFC) and Department of Internal Medicine I, University Hospital Würzburg, 97080 Würzburg, Germany; Gerull_B@ukw.de; 5Clinic for Thoracic and Cardiovascular Surgery, Heart and Diabetes Center NRW, University Hospital of the Ruhr–University Bochum, Georgstrasse 11, 32545 Bad Oeynhausen, Germany; 6Center for Congenital Heart Defects, Heart and Diabetes Center NRW, University Hospital of the Ruhr–University Bochum, Georgstrasse 11, 32545 Bad Oeynhausen, Germany; sschubert@hdz-nrw.de (S.S.); tlaser@hdz-nrw.de (K.T.L.); 7Medical Research and Educational Center, Lomonosov Moscow State University, Lomonosovsky Prospect 27, Building 10, 119991 Moscow, Russia; elena_mershina@mail.ru (E.M.); pilius.polina@botkin.ai (P.P.); vsi-ni@mail.ru (V.S.); 8National Medical Research Center for Cardiology, 3–ya Cherepkovskaya Street 15A, 121552 Moscow, Russia; prof.boytsov@gmail.com

**Keywords:** desmoglein-2, desmocollin-2, DSG2, DSC2, ARVC, ACM, LVNC, cardiomyopathy, desmosomes, desmin

## Abstract

About 50% of patients with arrhythmogenic cardiomyopathy (ACM) carry a pathogenic or likely pathogenic mutation in the desmosomal genes. However, there is a significant number of patients without positive familial anamnesis. Therefore, the molecular reasons for ACM in these patients are frequently unknown and a genetic contribution might be underestimated. Here, we used a next-generation sequencing (NGS) approach and in addition single nucleotide polymor-phism (SNP) arrays for the genetic analysis of two independent index patients without familial medical history. Of note, this genetic strategy revealed a homozygous splice site mutation (DSG2–c.378+1G>T) in the first patient and a nonsense mutation (DSG2–p.L772X) in combination with a large deletion in DSG2 in the second one. In conclusion, a recessive inheritance pattern is likely for both cases, which might contribute to the hidden medical history in both families. This is the first report about these novel loss-of-function mutations in DSG2 that have not been previously identi-fied. Therefore, we suggest performing deep genetic analyses using NGS in combination with SNP arrays also for ACM index patients without obvious familial medical history. In the future, this finding might has relevance for the genetic counseling of similar cases.

## 1. Introduction

Desmosomes are multiple protein complexes mediating the cell–cell adhesion of cardiomyocytes and epithelial cells [1]. They consist of proteins from three different protein families. The cadherins desmoglein-2 and desmocollin-2 (encoded by *DSG2* and *DSC2*) are type I transmembrane glycoproteins [2,3,4] and mediate the calcium-dependent cell–cell adhesion between cardiomyocytes. Both proteins have extracellular domains consisting of four cadherin domains (EC1–4) and an anchor domain (EA) [5]. Heterophilic protein–protein interactions between the desmosomal cadherins are mediated in trans by a strand-swap mechanism of their first EC domains [2]. Two proteins from the Armadillo family, which are called plakophilin-2 (*PKP2*) and plakoglobin (*JUP*), bind to the cytoplasmic domains of the desmosomal cadherins [6]. The Armadillo domain formed by varying numbers of Armadillo repeats is characteristic for these proteins [7,8]. Plakophilin-2 and plakoglobin bind to desmoplakin (*DSP*), which is a member of the plakin family connecting the desmosomes to the cardiac intermediate filaments, which are mainly built by desmin (*DES*) [9,10]. Of note, mutations in the cardiac desmosomal genes and in addition in *DES* cause arrhythmogenic cardiomyopathy (ACM, MIM, #609040) [11,12,13,14,15,16,17]. However, *DES* mutations beside ACM also cause other cardiomyopathies like dilated (DCM) [18], restrictive (RCM) [19,20], or left ventricular non-compaction cardiomyopathy (LVNC) [21,22]. In rare cases, mutations in non-desmosomal genes have also been described for ACM [23,24,25]. ACM is clinically characterized by ventricular arrhythmia in combination with right or biventricular dilation [26]. Originally, the term arrhythmogenic right ventricular cardiomyopathy (ARVC) was used and clinical task force guidelines have been developed and modified for the specific clinical diagnosis of ARVC [27]. However, in the end-stage phase of ARVC the left ventricle is also frequently affected [28]. Even dominant left ventricular forms have been described [29], indicating an overlap between arrhythmogenic DCM and ARVC. Despite precise clinical definitions of ACM, this term is therefore more frequently used in the newer literature [30]. Progressive replacement of the myocardium against fibrotic and fatty tissue at the cellular and histological level is characteristic for ACM [31,32]. It is estimated that 5–10% of cases with ACM carry a pathogenic *DSG2* mutation [33]. However, the majority of *DSG2* variants are rare missense variants with unknown significance [3,34] (http://www.hgmd.cf.ac.uk/ac/index.php, accessed on 20 January 2021). For some of the known *DSG2* missense mutations, cleavage defects of the pro-domain have been described [35]. In addition, a considerable number of heterozygous pathogenic loss-of-function mutations (LoF) like nonsense, insertions, deletions, or splice site mutations leading to frameshifts and premature termination codons (PTCs) were found in ACM patients [36,37,38].

Here, we describe two unrelated index patients (Figure 1) where we identified homo- or hemizygous *DSG2* LoF mutations by next-generation sequencing (NGS) leading to ACM. Single nucleotide polymorphism (SNP) arrays revealed consanguinity in the first family and a large additional *DSG2* deletion, spanning the region from intron 1 to 14 in the second index patient (*DSG2*–Ex2_Ex14del). Because no further family members were clinically affected, we suggest a recessive inheritance in both cases.

## 2. Results

### 2.1. Clinical Description of the Patients

#### 2.1.1. Family A

The index patient (II-1, Figure 1A) belongs to a South Asian family without obvious familiar history of cardiomyopathies. 

At the age of 12, he survived a syncope and received afterwards his clinical diagnosis of ARVC according to the revised task force criteria [27]. In detail, echocardiography and magnetic resonance imaging (MRI) revealed normal left ventricular structure (left ventricular end-diastolic diameter, LVEDD = 43 mm) and function (fractional shortening, FS = 33%, end-diastolic volume 78 mL/m^2^, left ventricular ejection fraction LVEF = 69%) at this time. However, the right ventricle was dilated (parasternal long axis right ventricular outflow view, PLAX RVOT = 33 mm in the long axis view, FAC 31%, end-diastolic volume 138 mL/m^2^, EF 29% with additional segmental bulging, Figure 2A,B). Analysis of a right ventricular biopsy revealed slight endo-fibrotic elastosis with proliferating myofibroblasts (Figure 2C,D). In addition, a right axis deviation, primarily incomplete, later complete right bundle branch block, non-sustained ventricular tachycardia and ventricular extra beats were present (Figure 2E,F) in the ECG. Speckle tracking echocardiography (STE) revealed dyssynchrony of the septum and right ventricular wall (Figure 3 and Appendix A). Therefore, the patient was supplied with an implantable cardioverter-defibrillator (ICD). Clinical follow-up using echocardiography resulted in an increasing dilatation of the right ventricle (RVEDD = 69 mm, 17 years) and decreasing right ventricular function (TAPSE = 14 mm) in combination with paradoxical septal motions and a dilated right atrium. Because of ventricular tachycardia, the patient received several ICD shocks. At the age of 22, an additional left ventricular systolic dysfunction (LVEF = 38%) without significant left ventricular dilation (LVEDD = 51 mm) was detected. Concentration of n-terminal pro-brain natriuretic peptide (NT-proBNP) was slightly increased (952 pg/mL) at this time. No further family members developed any significant cardiac phenotype.

#### 2.1.2. Family B

The index patient (II-2, Family B, Figure 1B) belongs to a Caucasian family without obvious familiar history of cardiomyopathies. At the age of 22, he was hospitalized with severe pain in his left thigh and acute arterial thrombosis was diagnosed. Afterwards, cardiac MRI and echocardiography were used to identify the possible cardiac reasons for acute thrombosis. These investigations revealed biventricular enlargement (LVEDD = 64 mm) and reduced systolic LV and RV function (LVEF = 37%, right ventricular ejection fraction RVEF = 23%, Figure 4A–F) and signs of the left-ventricular non-compaction morphology using criteria defined by Petersen and Grothoff et al. [39,40]. LVNC is mainly characterized by increased endomyocardial trabeculations. Diagnosis of ACM was based on the Padua criteria [41]. The patient fulfilled all major and minor criteria, excluding a family history. Holter monitoring ECG showed a sinus rhythm with frequent premature ventricular beats and non-sustained ventricular tachycardia (Figure 4J,K). Therefore, an ICD was implanted. Several follow-ups were held afterwards and revealed a progressive decrease of the left ventricular systolic function (LVEF = 21%) and enlarged end-diastole LV size (LVEDD = 65 mm). Due to progressive ARVC, he was consulted for heart transplantation. Cardiologists clinically examined all other family members (Figure 1B). Cardiac MRI revealed normal heart structure and function. Symptoms of ARVC were absent in I-1, I-2, II-1, and II-3 (Figure 1B). However, cardiac MRI of the proband’s mother (I-2, Family B) revealed left-ventricular non-compaction morphology according to criteria of Petersen et al. [39] (Figure 5B).

### 2.2. Genetic Analyses

Index patient II-1 of Family A (Figure 1A) developed a progressive ACM. Although no further family members were clinically affected, we performed a genetic analysis of the affected index patient using a broad NGS panel, covering 174 genes associated with cardiomyopathies or syndromes with cardiac involvement. Filtering using a MAF < 0.001 revealed six heterozygous missense variants and one homozygous mutation in the donor splice site of *DSG2* exon 4 (*DSG2*–c.378+1G>T, Table 1, Figure 6A). Sanger sequencing was used for verification of *DSG2*–c.378+1G>T (Figure 6B). *DSG2* is localized on chromosome 18 and encodes desmoglein-2. In theory, three different reasons might explain homozygosity of *DSG2*–c.378+1G>T.
Consanguinity of the parentsAn additional large deletion in *DSG2* localized on the second chromosome mimicking homozygosity *DSG2*–c.378+1G>T orUniparental isodisomy (UPD)

Since no genomic DNA of the parents or siblings was available, we used a SNP microarray for chromosomal analysis of II-1 (Family A, Figure 6C). A large putative deletion on chromosome 18 was excluded by this analysis (Figure 6D). However, we identified a loss of heterozygosity (LOH) on chromosome 18 (15,605 kbp, Figure 6D). In addition, we found seven further regions with LOHs localized on autosomes (LOH > 5000 kb, Figure 6C). The total size of the autosomal LOH regions is 4.39% (total autosome LOH = 121,988 kbp; covered autosome length 2,781,797 kbp), supporting consanguinity of the parents. This fits to a parental fourth-degree relationship like, e.g., first cousins once removed (theoretical Froh = 3.125%, [42]). Because of these findings it can be suggested that both parents are obligate carriers for *DSG2*–c.378+1G>T leading to homozygosity of this mutation in II-1 (Figure 1A).

In family B with Russian origin, only the index patient (II-2, Figure 1B) received an ACM diagnosis. Comparable to family A, we started the genetic analysis with an NGS approach for the index patient II-2, revealing eight rare variants with a MAF < 0.001 (Table 2). Interestingly, comparable to family A we found exclusively reads for a nonsense mutation *DSG2*-p.L772X, indicating homo- or hemizygosity (Figure 7A). This finding was verified by Sanger sequencing. In addition, we genotyped the unaffected parents (I-1 and I-2) and the two siblings (II-1 and II-3). Of note, beside the heterozygous *DSG2*-p.L772X mutation, the father carried a further heterozygous SNP (*DSG2*-p.R773K, MAF = 0.2604, https://gnomad.broadinstitute.org/variant/18-29122799-G-A?dataset=gnomad_r2_1 (accessed on 20 January 2021), Figure 7B). Surprisingly, the mother (I-2) was wild type for both variants excluding parental consanguinity (Figure 7C). The older brother II‑1 of the index patient did not carry the nonsense variant, but carried the SNP *DSG2*-p.R773K in a heterozygous status (Figure 7D). The dizygotic twin sister II‑3 presented the same SNP in a homozygous status (Figure 7F). Since all children carried either *DSG2*-p.L772X or -p.R773K, it can be concluded that the father (I-1) carried both *DSG2* variants in a compound heterozygous status. To investigate the genetic reason for homozygosity of *DSG2*-p.L772X in II-2 and of *DSG2*-p.R773K in II-3, we performed a SNP microarray for the index patient II-2 and his parents (Appendix A). These analyses revealed a large additional deletion mutation affecting nearly the complete *DSG2* gene (Figure 7G–I) for the index patient (II-2) and his mother (I-2). In summary, we identified a hemizygous nonsense variant (*DSG2*-p.L772X) for the index patient and suggest therefore a recessive inheritance in Family B (Figure 1B) caused by paternal inheritance of *DSG2*-p.L772X and maternal inheritance of *DSG2*_Ex2-14del.

## 3. Discussion

In 2002, Eshkind et al. demonstrated that the global homozygous knock-out of *Dsg2* in mice is embryonic lethal [43]. Therefore, different conditional knock-out, knock-in, and transgenic mouse models for *Dsg2* leading to murine arrhythmogenic cardiomyopathies have been developed [44,45,46,47,48,49]. In spite of this, the first pathogenic *DSG2* mutations associated with ACM in humans were found in 2006 [37,50]. Currently, more than 200 different *DSG2* variants associated with ACM or DCM are listed in the Human Gene Mutation Database (HGMD, http://www.hgmd.cf.ac.uk/ (accessed on 20 January 2021), Qiagen, Hilden, Germany). The majority of them are rare missense variants with unknown significance and/or with unknown pathomechanism. However, small insertions, deletions, indels, and splicing mutations in *DSG2* have been also described. Presumably, these LoF mutations in *DSG2* might induce nonsense-mediated mRNA decay and in consequence lead to loss of the adhesive function of the desmosomes as suggested by Kant et al. using heart-specific *Dsg2*-deficient mice [46]. Interestingly, recessive homozygous or compound heterozygous *DSG2* mutations are rare, indicating a dominant inheritance in most cases [51]. In this report, we describe two independent index patients with ACM carrying either a homozygous splice site mutation in *DSG2* or a hemizygous nonsense mutation. Although we cannot completely exclude modifying effects of rare variants of unknown significance (VUS) like, e.g., *PKP2*-p.R811S in II-1, Family A (Table 1), it is unlikely that these VUS are primary causative based on their MAF. Mutations in *PKP2* are common in ARVC [11]. However, most of them are truncating mutations leading to haploinsufficiency.

Homozygosity is caused in Family A by consanguinity and hemizygosity is caused in Family B by a large deletion on the second chromosome. In Family B the index patient received the nonsense mutation from the father and the large deletion from the mother. Interestingly, no further family members developed clinical symptoms of ACM. Therefore, for both presented cases it can be suggested that the ACM phenotype is caused by a recessive inheritance of *DSG2* LoF mutations. Uniparental isodisomy, as we have recently described for the homozygous mutation *DSC2*-c.1913_1916delAGAA leading to the truncation and degradation of desmocollin-2 [52], can be excluded by SNP microarray analyses in both cases. Recently, some other homozygous *DSC2* mutations associated with ACM were described, indicating a recessive inheritance in specific cases for this gene [53,54,55,56].

In general, for about 50% of ACM patients a pathogenic mutation in the desmosomal genes can be identified [33]. Rare de novo mutations in isolated index patients [57] and rare copy number variants [58,59] might explain some cases without obvious familial/genetic history. In addition, we show here two independent cases with recessive *DSG2* mutations, which can contribute to a hidden family anamnesis. It is worth noting that in proband (II-2, Family B) and his mother (I‑2, Family B) a left ventricular non-compaction morphology was detected according to echocardiography and MRI analyses [39]. However, it does not meet the modern MRI criteria for non-compacted myocardium [40]. Currently, we cannot exclude that a mutation in *DSG2* might modify this pathology. It is noteworthy that the disease in proband debuted with the development of acute arterial thrombosis, most likely cardio-embolic origin on the background of a non-compaction in the LV.

Despite the recessive type of inheritance in both families (A and B), we suggest to continue dynamic clinical investigations of heterozygous mutation carriers in the future as long as possible because we cannot exclude that a cardiac phenotype will occur at a later onset.

## 4. Materials and Methods

An NGS panel covering 174 cardiomyopathy-associated genes (TrueSight Cardio Panel, Illumina, San Diego, CA, USA, Appendix B) was used for sequencing of the index patient in Family A (II-1, Family A). A minor allele frequency (MAF) < 0.001 (The Genome Aggregation Database, gnomAD, 19 January 2021) was applied for filtering relevant variants. Sanger sequencing was used for verification of *DSG2*–c.378+1G>T (Macrogen, Amsterdam, Netherlands). A SNP array, CytoScan HD (Affimetrix, Santa Clara, CA, USA), was performed using the genomic DNA from II-1 (Family A) by Atlas Biolabs (Berlin, Germany). The overall average marker spacing of the CytoScan HD is 1148 base pairs (intragenic = 880 and intergenic = 1737). Chromosome Analysis Suite V4.2.0.80 (Thermo Fisher Scientific, Waltham, USA) was used for analysis of SNP arrays. No DNA samples of further family members were available for genetic analyses in this family. For histology of the right ventricular biopsy from II-1, Family A, a hematoxylin and eosin staining (HE) and a trichrome staining (TC) was done using standard protocols [32].

An NGS exome sequencing analysis using Nextseq 550 (Illumina, San Diego, CA, USA) was done for the index patient in Family B (II-2, Figure 2B). Libraries were prepared using the Truseq DNA Library Preparation Kit (Illumina, San Diego, CA, USA) and the xGen Exome Research Panel (IDT, Integrated DNA Technologies, Coralville, IA, USA). Variants with MAF < 0.001 in 188 cardiomyopathy-associated genes (Appendix C) were analyzed. The verification of DSG2–p.L772X and DSG2–Ex2_Ex14del was done by Sanger sequencing on an Applied Biosystems 3500 DNA Analyzer (Thermo Fisher Scientific, Waltham, MA, USA). All stages of sequencing were carried out according to the manufacturers’ protocols. CytoScan HD arrays (Affimetrix, Santa Clara, CA, USA) were used for I‑1, I-2, and II-1 (Family B) and were performed by Atlas Biolabs (Berlin, Germany). Analysis of SNP arrays was done using Chromosome Analysis Suite V4.2.0.80 (Thermo Fisher Scientific, Waltham, USA).

## 5. Conclusions

Our genetic analyses of two independent ACM index patients without obvious familial anamnesis revealed homo- or hemizygous LoF mutations in *DSG2*. Therefore, we suggest also for ACM patients without further affected family members a genetic counseling and analysis, because putative pathogenic mutations might be hidden by a recessive inheritance.

## Figures and Tables

**Figure 1 ijms-22-03786-f001:**
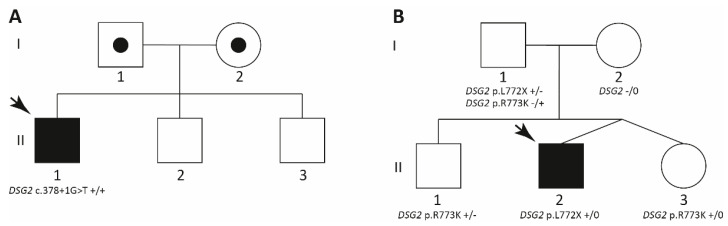
Pedigrees of the described families. (**A**) Family A has a South Asian origin. The male index patient (II-1) received his diagnosis of arrhythmogenic right ventricular cardiomyopathy (ARVC) at the age of 12. (**B**) Family B has a Russian origin. Circles represent females, squares males. Black-filled symbols indicate a cardiac phenotype and white symbols indicate healthy family members. +/− indicates heterozygous, +/+ homozygous, and +/0 hemizygous status. Index patients are marked with an arrow. Obligate carriers are shown with a black dot in the pedigree symbol.

**Figure 2 ijms-22-03786-f002:**
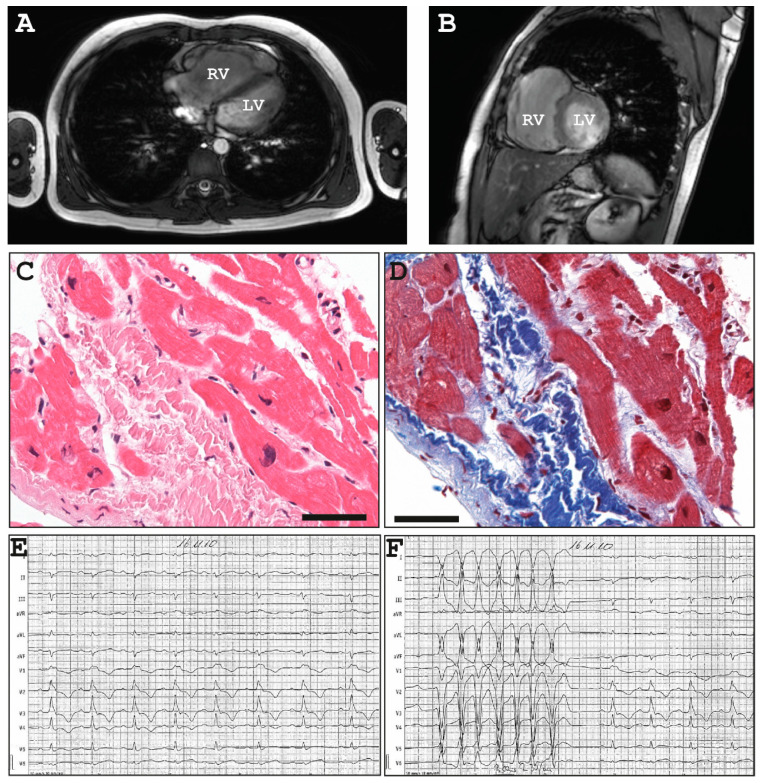
Clinical data of II-I, Family A. (**A–B**) Cardiac magnetic resonance imaging (MRI) revealed a dilated right ventricle with reduced global function and segmental bulging. Hematoxylin and eosin (**C**) and trichrome staining (**D**) of a right-ventricular biopsy revealed subendocardial fibrosis and degeneration of cardiomyocytes. Scale bars represent 50 µm. (**E–F**) Right axis deviation, at the age of 12 years still incomplete, later complete right bundle branch block, inverted T-waves in the right precordial leads and non-sustained ventricular tachycardia were present in the 12-lead electrocardiogram (ECG).

**Figure 3 ijms-22-03786-f003:**
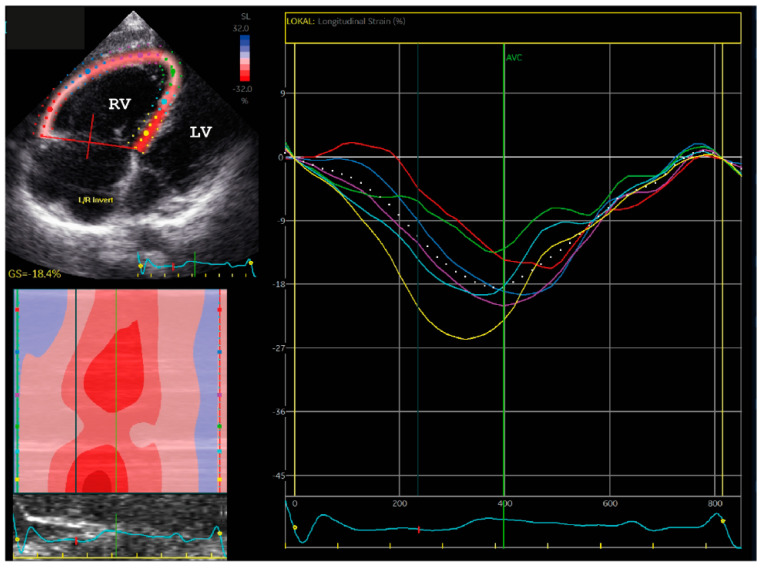
Speckle tracking echocardiography (STE) demonstrated dyssynchrony of the septal and right ventricular free-wall segments (see also Appendix A).

**Figure 4 ijms-22-03786-f004:**
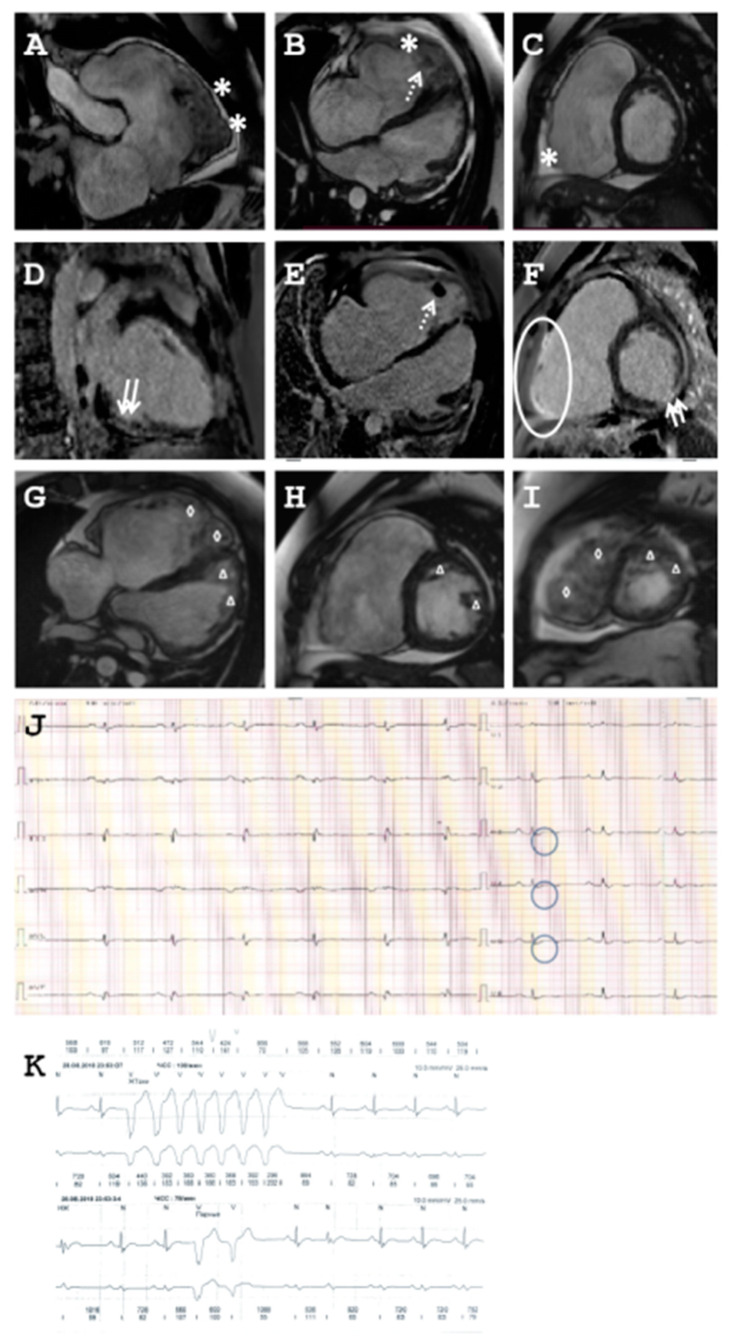
Clinical data of Family B. (**A–I**) Cardiac magnetic resonance imaging (MRI) of index patient (II-2, Family B). Asterisk indicates thinning and aneurysmal protrusion of right ventricular wall (**A–C**). Arrows indicate ischemic changes in left ventricle (**D,F**). Dotted arrows indicate thrombosis in the right ventricle (**B,E**). Ellipse shows fibrosis of the right ventricle (**F**). Triangle demonstrates LV non-compaction (**G–I**). Rhombus demonstrates RV non-compaction (**G,I**). (**J**) Epsilon waves were present in the 12-lead ECG. (**K**) Tachycardia was present in the Holter monitoring-ECG.

**Figure 5 ijms-22-03786-f005:**
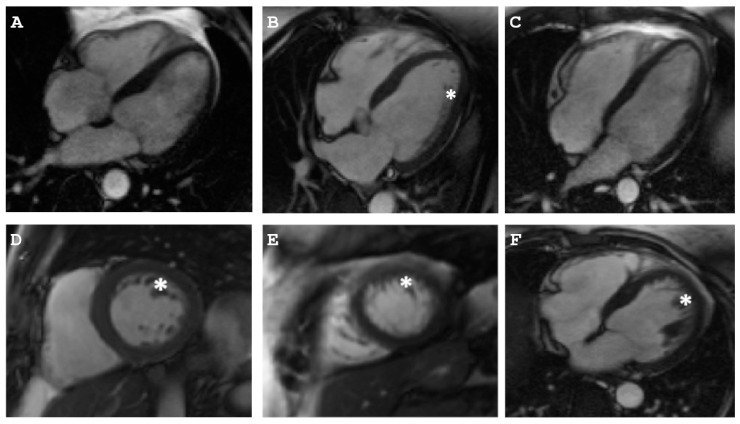
Cardiac magnetic resonance imaging (MRI) of the I-1 (**A**), I-2 (**B,D–F**), and II -1 (**C**) of Family B. Asterisk indicates non-compaction layer.

**Figure 6 ijms-22-03786-f006:**
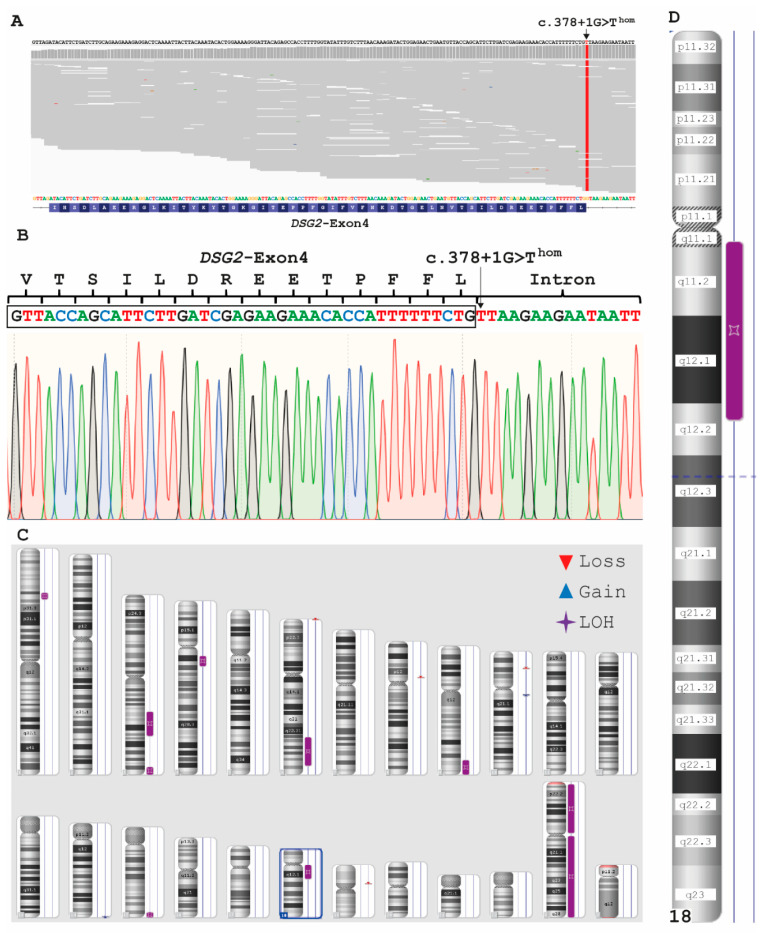
Genetic analysis of II-1 (Family A). (**A**) Integrated genome view of exon 4 in *DSG2* of patient II-1. (**B**) Electropherogram of exon 4 in *DSG2* (II-1). The mutation DSG2–c.378+1G>T is affecting the donor splice site and is found in a homozygous status. (**C**) Karyoview of II-1. LOH = loss of heterozygosity. (**D**) Detailed ideogram of human chromosome 18 revealing a 15.6 Mb loss of heterozygosity (LOH).

**Figure 7 ijms-22-03786-f007:**
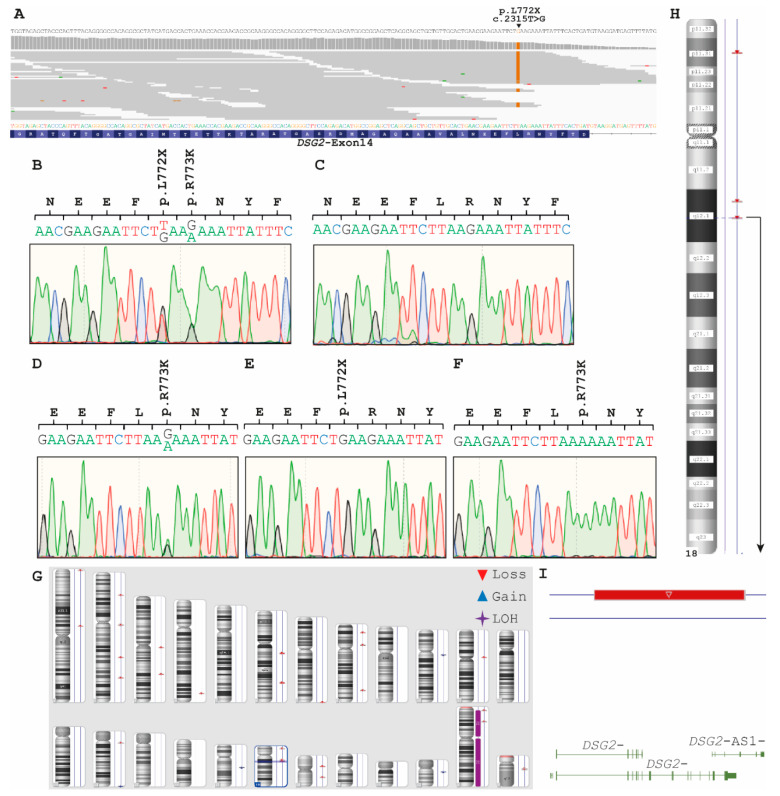
(**A–I**) Genetic analysis of Family B. (**A**) Integrated genome view of exon 14 in *DSG2* of patient II-2 (Family B). (**B–F**) Partial electropherograms of *DSG2* exon 14 revealed for I-1 two heterozygous variants p.L772X and p.R773K (**B**), for I‑2 a wild-type sequence (**C**), for II-1 the heterozygous variant p.R773K (**D**), for II-2 hemizygous p.L772X (**E**), and for II-3 hemizygous p.R773K (**F**). (**G**) Karyoview of II-2 (Family B). LOH = loss of heterozygosity. (**H–I**) Detailed ideogram of human chromosome 18 revealing an additional deletion on the second chromosome affecting nearly the complete *DSG2* gene.

**Table 1 ijms-22-03786-t001:** List of rare variants (MAF < 0.001) identified in the index patient II-1 (Family A).

Gene	Genomic Coordinates	Transcript	Kind of Mutation	Protein Change	MAF ^1^	ACMG Classification
*DSG2*	18:29100928	NM_001943.3	Splice Site Mutation	Unknown	novel	Likely pathogenic
*TBX3*	12:115120963	NM_016569.3	Missense	p.M15V	novel	VUS
*TRIM63*	1:26380423	NM_032588.3	Missense	p.D338Y	0.0002671	VUS
*PKP2*	12:32949101	NM_004572.3	Missense	p.R811S	0.00002828	VUS
*SDHA*	5:236628	NM_004168.2	Missense	p.A449V	0.000003543	VUS
*TTN*	2:179650454	NM_001267550.1	Missense	p.D796N	0.00007782	VUS
*LTBP2*	14:74975348	NM_000428.2	Missense	p.A1204V	0.0008008	VUS

^1^ According to the Genome Aggregation Database (gnomAD), https://gnomad.broadinstitute.org/, 20 January 2021; ACMG = American College of Medical Genetics and Genomics, MAF = minor allele frequency, VUS = variant of unknown significance.

**Table 2 ijms-22-03786-t002:** List of rare variants (MAF < 0.001) identified in the index patient II-2 (Family B).

Gene	Genomic Coordinates	Transcript	Kind of Mutation	Protein or cDNA Change	MAF ^1^	ACMG Classification
*DSG2*	18:29122796	NM_001943.5	nonsense	p.L772X	Novel	Likely pathogenic
*TTN*	2:179497960	NM_001256850.1	missense	p.V12706A	0.000008059	VUS
*PRDM16*	1:3331216	NM_022114.3	unknown	c.2691+5G>A	0.0002009	Likely benign
*VCL*	10:75849080	NM_014000.3	missense	p.S383R	Novel	VUS
*SYNE2*	14:64545208	NM_182914.2	missense	p.S3683T	0.0002230	VUS
*BAG3*	10:121431767	NM_004281.4	missense	p.R170W	0.000004054	VUS
*PRDM16*	1:3328948	NM_022114.4	missense	p.F729L	0.0003899	Likely benign
*RYR1*	19:39014565	NM_000540.3	missense	p.I3484T	0.00005576	VUS

^1^ According to the Genome Aggregation Database (gnomAD), https://gnomad.broadinstitute.org/, 20 January 2021; ACMG = American College of Medical Genetics and Genomics, MAF = minor allele frequency, NA = not applicable, VUS = variant of unknown significance.

## Data Availability

The data used and/or analyzed during the current study are available from the corresponding authors on reasonable request.

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
