# Peer review of "Hemi- and Homozygous Loss-of-Function Mutations in DSG2 (Desmoglein-2) Cause Recessive Arrhythmogenic Cardiomyopathy with an Early Onset"

_ijms, 2021, doi:10.3390/ijms22073786_

Round 1

Reviewer 1 Report

The authors used a next generation sequencing (NGS) approach and additionally single  nucleotide polymorphism (SNP) arrays for the genetic analysis of two  independent index patients suffering from Arrhythmogenic Cardiomyopathy (ACM) without familial medical history. Of note, this genetic  strategy revealed a homozygous splice site mutation (DSG2-c.378+1G>T) in the  first patient and a nonsense mutation (DSG2-p.L772X) in combination with a large deletion in DSG2 in the second one. They concluded that a recessive inheritance pattern is likely for both cases, which might contribute to the hidden medical history in both families and suggested performing deep genetic analyses using NGS in combination with SNP arrays also for ACM index patients without obvious familial medical history.  

Comments/queries

Phenotypes describe a set of identifiable clinical characteristics that help to distinguish affected individuals from unaffected. The term ACM represents an evolution of diagnostic terms describing disease of the LV and RV, but it was only recently that the term was formally used to describe this disorder. Diagnostic criteria for the expanded ACM spectrum do not yet exist (Protonotarios A, Elliott PM. Arrhythmogenic Cardiomyopathy: A Disease or Merely a Phenotype? Eur Cardiol. 2020. PMID: 32180838). On which criteria was the diagnosis of ACM based? If I argue that these two patients based on the differences in clinical manifestations and genotype suffered from different diseases how would the authors respond? These two points should  be thoroughly discussed and clarified in the manuscript.

Author Response

First, we want to thank reviewer #1 for the constructive suggestions to our manuscript. We agree with reviewer #1 that the term arrhythmogenic cardiomyopathy (ACM) is more frequently used in the newer literature, whereas the term arrhythmogenic right ventricular cardiomyopathy (ARVC) was used more frequently in older literature. However, it was recognized and well established that in the end-stage of this disease also the left ventricle is involved [1]. In addition, there are also some reports, which describe cases with arrhythmogenic left-ventricular cardiomyopathy (ALVC) [2]. We agree with reviewer #1 that for ARVC there are defined task force criteria available [3] whereas the term ACM is unfortunately not 100% precisely defined. Therefore, we indicate in the revised manuscript the clinical diagnosis of ARVC according to the task force criteria [3,4]. ACM in the index patient from family B was diagnosed by Padua criteria [4] (5 major for ARVC and 1 major ALVC). For right ventricle: 1) Major: Regional RV akinesia, dyskinesia with global RV dilatation and global RV systolic dysfunction; Fibrous replacement of the myocardium in ≥ 1 segments, with or without fatty tissue ; Inverted T waves in right precordial leads (V1,V2, and V3); Frequent ventricular extra systoles (>500 per 24 hours), non-sustained or sustained ventricular tachycardia of LBBB morphology; 2) Minor: Epsilon wave. For left ventricle: 1) Major: fibrosis of≥1 segments of the free wall (subepicardial or midmyocardial); 2) Minor: Global LV systolic dysfunction; Low QRS voltages (<0.5 mV peak to peak ) in limb leads.

In addition, we describe in the introduction of our revised manuscript as suggested by reviewer #1 the different terms ACM and ARVC and explain their origin in more detail.

Reviewer 2 Report

The authors present a work relevant to the field, demonstrating with their findings the importance of Desmoglein 2 in cardiomyopathy. 
I recommend to improve the quality and reduce as far as possible the “noise” of the image which makes difficult to follow the signal in the ECGs of figure 2.

Author Response

Thanks for the comment of reviewer #2. Unfortunately, a better image quality is not available. They are the best we could get after ten years.

Reviewer 3 Report

Overall Comments: This a well-designed study and well-written manuscript detailing the clinical presentation and the genetic underpinnings of two “novel” mutations in probands presenting with arrhythmogenic cardiomyopathy. A next generation sequencing approach and SNP arrays were utilized of assess the genetic cause of disease in two independent index patients.  Variants in the desmosomal gene desmoglein-2 were identified in both patients, the first patient with a homozygous splice site mutatation DSG2-c378+1G>T) and the second patient with a nonsense mutation (DSG2-p.L772X) combined with a large gene deletion. Additional rare variants were identified in both index patient, that were not considered likely contributors to the presentation of disease in the patients. Additional findings include that parents of the first patient were both obligate carriers and were likely consanguineous. The second patient exhibited additional signs of left-ventricular non compaction similar to that of his mother. The genetic cause of the LVNC in this patient and his mother was not ascribed.

Major Concerns:

  1. It should be written more clearly in the abstract or introduction that this is the first report of these novel loss of function mutations that have not been previously identified.
  2. There should be a little more discussion about the potential that the other rare variants identified could affect disease presentation in both of the patients, the lists appear to include genes with known effects in the heart if mutated/absent. 
  3. Provide slightly more detail about the reporting of variants in Tables 1 and 2, were any of these found in any patients with cardiac manifestations?
  4. For the patient from Family B fibrosis is shown via the MRI image, likely due to lack of biopsied tissue. Can you show an image that provides visual evidence of LVNC in the patient?
  5. What do the authors expect is the impact of these particular DSG2 mutations and deletion on the normal function of desmosomes?

Minor Concerns:

  1. On Line 204, include the word left in front of the words “….systolic function.”
  2. There are small typos, though the writing is very good. E.g. In the last line of the abstract Line 56, insert the word “the” in front of “future” and Line78, replace “build” for “built”
  3. On line 158, explain why eosinophilic myocarditis and cardiac amyloidosis were excluded.
  4. On Line 199, it would be helpful for the more general reader if left ventricular non-compaction morphology was briefly described.
  5. In the introduction, provide the statistics of what percent of ARVC is caused by pathogenic mutations in DSG2 and what percent are considered rare missense variants with unknown significance? I see you have some of this information on Line 313, but it would be interesting to readers to know the incidence of rare pathogenic DSG2 ARVC mutations.
  6. Patient 2 has a number of confounding presentations, to what are you attributing the arterial thrombosis? How many genes associated with LVNC were screened?

Author Response

Reviewer 3

Overall Comments: This a well-designed study and well-written manuscript detailing the clinical presentation and the genetic underpinnings of two “novel” mutations in probands presenting with arrhythmogenic cardiomyopathy. A next generation sequencing approach and SNP arrays were utilized of assess the genetic cause of disease in two independent index patients. Variants in the desmosomal gene desmoglein-2 were identified in both patients, the first patient with a homozygous splice site muta[ta]tion DSG2-c378+1G>T) and the second patient with a nonsense mutation (DSG2-p.L772X) combined with a large gene deletion. Additional rare variants were identified in both index patient[s], that were not considered likely contributors to the presentation of disease in the patients. Additional findings include that parents of the first patient were both obligate carriers and were likely consanguineous. The second patient exhibited additional signs of left-ventricular non compaction similar to that of his mother. The genetic cause of the LVNC in this patient and his mother was not ascribed.

Answer of the authors

We thank reviewer #3 for this motivating statements to our manuscript.

Reviewer 3-1

It should be written more clearly in the abstract or introduction that this is the first report of these novel loss of function mutations that have not been previously identified.

Answer of the authors

We thank reviewer #3 for this suggestion and changed the abstract in the revised manuscript accordingly.

Reviewer 3-2

There should be a little more discussion about the potential that the other rare variants identified could affect disease presentation in both of the patients, the lists appear to include genes with known effects in the heart if mutated/absent. Provide slightly more detail about the reporting of variants in Tables 1 and 2, were any of these found in any patients with cardiac manifestations?

Answer of the authors

We discussed that these variants of unknown significance can be excluded as causative based on their minor allele frequency. However, we cannot completely exclude modifying influence on the phenotype. For example we found a missense variant PKP2-p.R811S in the genome of II-1, family A (Table 1). PKP2 mutations are common in ARVC [5]. However, the majority of pathogenic PKP2 mutations are truncating mutations leading to haploinsufficiency. We discussed this point in the revised manuscript.

Reviewer 3-3

For the patient from Family B fibrosis is shown via the MRI image, likely due to lack of biopsied tissue. Can you show an image that provides visual evidence of LVNC in the patient?

Answer of the authors

We have added changes to MRI images of Family B according to the suggestion of reviewer #3.

Reviewer 3-4

What do the authors expect is the impact of these particular DSG2 mutations and deletion on the normal function of desmosomes?

Answer of the authors

We thank reviewer #3 for this suggestion and discussed this point in more detail in the revised manuscript. Kant et al. used heart specific Dsg2 deficient mice and demonstrated ultrastructural defects of the intercalated disc [6]. Therefore, the authors suggested a loss of adhesive function as the main pathomechanisms [6].

Reviewer 3-5

On Line 204, include the word left in front of the words “….systolic function.”

There are small typos, though the writing is very good. E.g. In the last line of the abstract Line 56, insert the word “the” in front of “future” and Line78, replace “build” for “built”

Answer of the authors

We thank reviewer #3 for these suggestions and changed it accordingly.

Reviewer 3-6

On line 158, explain why eosinophilic myocarditis and cardiac amyloidosis were excluded.

Answer of the authors

We removed this sentence in the revised version of the manuscript to prevent over interpretation of this histology.

Reviewer 3-7

On Line 199, it would be helpful for the more general reader if left ventricular non-compaction morphology was briefly described.

Answer of the authors

We thank reviewer #3 for this suggestion and explained that increased endomyocardial trabeculations is characteristic for LVNC.

Reviewer 3-8

In the introduction, provide the statistics of what percent of ARVC is caused by pathogenic mutations in DSG2 and what percent are considered rare missense variants with unknown significance? I see you have some of this information on Line 313, but it would be interesting to readers to know the incidence of rare pathogenic DSG2 ARVC mutations.

Answer of the authors

We include the requested information in the introduction of the revised manuscript.

Reviewer 3-9

Patient 2 has a number of confounding presentations, to what are you attributing the arterial thrombosis? How many genes associated with LVNC were screened?

Answer of the authors

We associated arterial thrombosis with the presence of a non-compacted layer in the left ventricle. Other causes of arterial thrombosis were excluded: Antiphospholipid syndrome and hereditary thrombophilia. We added changes to description of genes in Appendix B, Genes associated with LVNC are separated from genes associated with other cardiomyopathies. All genes, which were screened in the genome of this patient, are indicated in Appendix B of the revised manuscript.

Additional Literature

  1. Sen-Chowdhry, S.; Syrris, P.; Ward, D.; Asimaki, A.; Sevdalis, E.; McKenna, W.J. Clinical and genetic characterization of families with arrhythmogenic right ventricular dysplasia/cardiomyopathy provides novel insights into patterns of disease expression. Circulation 2007, 115, 1710-1720, doi:10.1161/CIRCULATIONAHA.106.660241.
  2. Bermudez-Jimenez, F.J.; Carriel, V.; Brodehl, A.; Alaminos, M.; Campos, A.; Schirmer, I.; Milting, H.; Abril, B.A.; Alvarez, M.; Lopez-Fernandez, S., et al. Novel Desmin Mutation p.Glu401Asp Impairs Filament Formation, Disrupts Cell Membrane Integrity, and Causes Severe Arrhythmogenic Left Ventricular Cardiomyopathy/Dysplasia. Circulation 2018, 137, 1595-1610, doi:10.1161/CIRCULATIONAHA.117.028719.
  3. Marcus, F.I.; McKenna, W.J.; Sherrill, D.; Basso, C.; Bauce, B.; Bluemke, D.A.; Calkins, H.; Corrado, D.; Cox, M.G.; Daubert, J.P., et al. Diagnosis of arrhythmogenic right ventricular cardiomyopathy/dysplasia: proposed modification of the Task Force Criteria. Eur Heart J 2010, 31, 806-814, doi:10.1093/eurheartj/ehq025.
  4. Corrado, D.; Perazzolo Marra, M.; Zorzi, A.; Beffagna, G.; Cipriani, A.; Lazzari, M.; Migliore, F.; Pilichou, K.; Rampazzo, A.; Rigato, I., et al. Diagnosis of arrhythmogenic cardiomyopathy: The Padua criteria. Int J Cardiol 2020, 319, 106-114, doi:10.1016/j.ijcard.2020.06.005.
  5. Gerull, B.; Heuser, A.; Wichter, T.; Paul, M.; Basson, C.T.; McDermott, D.A.; Lerman, B.B.; Markowitz, S.M.; Ellinor, P.T.; MacRae, C.A., et al. Mutations in the desmosomal protein plakophilin-2 are common in arrhythmogenic right ventricular cardiomyopathy. Nat Genet 2004, 36, 1162-1164, doi:10.1038/ng1461.
  6. Kant, S.; Krull, P.; Eisner, S.; Leube, R.E.; Krusche, C.A. Histological and ultrastructural abnormalities in murine desmoglein 2-mutant hearts. Cell Tissue Res 2012, 348, 249-259, doi:10.1007/s00441-011-1322-3.